# MoE-ERAS : Expert Residency Aware Scheduling

Abhimanyu Bambhaniya [†], Sashankh Chengavalli Kumar [†], Tushar Krishna [†],

[†]Georgia Institute of Technology.

*abambhaniya3@gatech.edu, cksash@gatech.edu, tushar@ece.gatech.edu*

### Abstract

Mixture of Experts models have quickly grown in popularity due to their faster inference and training than dense models of similar capability. Parameter compression and offloading allows the users to run these model on smaller GPU memory (leading to cost savings). However, unpredictability in expert activation results in slower inference for offloaded experts. In this work, we profile and study the expert activation patterns when running large MoE models. Based on insights from activation patterns, we propose a new way of expert selection, which takes the expert residency into account. We introduce *MoE-ERAS*, Expert Residency Aware Selection to select the most suitable experts considering **both performance and accuracy**. We show substantial gains in decoding latency and expert swaps, and present analysis to show pre-fetching opportunities for future work. MoE-ERAS allows users to choose an acceptable point on the speedup-quality trade-off.

## I. Introduction

Numerous recent advancements in natural language processing hinge on large pre-trained language models, exemplified by entities like GPT-3 and 4 [1], [4], Palm & Gemini [5], [7], among others. However, the swift strides in this domain owe much to openly accessible LLMs like LLaMA [17], Mixtral [12], OPT [21], and many more. The primary boon of these open-access LLMs lies in researchers' ability to deploy and tailor them locally - a feat infeasible with proprietary APIs. With LLM models growing at an unprecedented scale, a Mixture of expert(MoE) models is one of the most promising directions that help us scale models to larger dimensions. The scale of a model is one of the most important axes for better model quality. Given a fixed computing budget, it is generally considered better to train a larger model for fewer steps, than to train a smaller model for more steps.

Despite the openly available range of LLMs, their sizes poses a challenge to utilization. **Cutting-edge open-access language models demand multiple high-end GPUs even for rudimentary inference tasks.** To make these LLMs feasible on more economical hardware setups, practitioners must either reduce model parameters or transfer parameters to *less expensive but slower storage* mediums, whether RAM or SSD [2], [14].

When moving the parameters to slower storage, the challenge of unreliable expert activation during the inference stage has emerged as a crucial issue in the development and optimization of MoE models. The gating mechanism determines which expert to activate for a given input just before the expert layer. This leads to sub-optimal model throughput as the appropriate weights would have to be brought from the host to the device, if not already on-chip.

We aim to increase the reuseblity of experts that are already present on the HBM rather than bring new experts each time from host memory. **Our key insight is that activating good enough experts can help in running the models faster and could also help us gain significant throughput improvements.** Building on this insight we introduce, MoE-ERAS, Expert Residency Aware Selection. MoE-ERAS is an expert selection method that takes the expert residency factor into consideration when selecting experts for each token. We propose 2 techniques, thresholding and biasing to tweak the router (gating network) for selecting throughput favourable experts.

Our approach is orthogonal to past works which aim to speedup MoE inference through quantization, prefetching and cacheing. To prove this, we implement MoE-ERAS on top of techniques like quantization and caching, and still show significant speedups with minimal accuracy degradation.

Our contributions in this work are :

- Profile and analyze the expert activation patterns for Mixtral-8x7B and Switch Transformer-32E. We collect over 500k token samples and provide insights for expert activation profiles.
- We introduce MoE-ERAS, Expert Residency Aware Selection, a smarter expert selection technique which factors in the locality (HBM or host) of expert during expert activation. Using MoE-ERAS, we see upto 21.2% reduction in the inference latency on top of other techniques like LRU caching and quantization.
- We present a speedup-quality trade-off with different techniques of MoE-ERAS. We evaluated the accuracy of different techniques on Wikitext2, C4 and MMLU.

## II. Motivation and Background

Mixture-of-experts models have emerged as a powerful approach for scaling up deep learning models to handle complex tasks with high-dimensional data. By dividing the computational workload among multiple expert sub-networks, each specializing in a different aspect of the input, MoE models can achieve high representational capacity while maintaining computational efficiency during training and inference. However, despite their promising performance, MoE models still face challenges regarding inference time, which can significantly impact their practical deployment, especially in latency-sensitive applications.

## A. Gating in Mixture-of-Experts

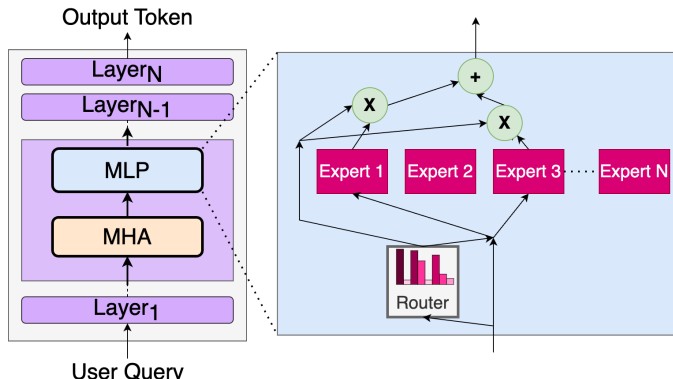

Fig. 1: The gating network of a mixture-of-experts model.

In mixture-of-experts models, the gating mechanism is crucial in selectively activating the appropriate expert sub-networks for a given input. The gating network is a separate component that takes the input data and produces a set of soft assignment scores or gating values, one for each expert. An example of gating is shown in 1. These gating values represent each expert's relative importance or suitability in handling the input data. The gating network is typically a single, fully connected network trained jointly with the experts during the model's training phase.

The gating values are then used to compute a weighted combination of the outputs from the individual selected experts, effectively creating a mixture of expert predictions. The gating mechanism allows the MoE model to dynamically allocate computational resources to the most relevant experts, enabling efficient processing of diverse input data while maintaining high predictive performance.

### B. Serving mixture of expert models

MoE serving can be divided into 2 baskets: either we can have enough devices (GPUs, TPUs, etc) to store all parameters (weights+KV Cache) inside on-chip HBM memory, or we have fewer devices and offload some of the unused parameters to the slower host memory(CPU DRAM). For the first approach, more devices are required; thus, we would have a higher cost of serving.

The second approach and main use case, which our work targets, is to use fewer devices and offload some parameters to the host memory, resulting in cheaper inference. However, offloading the experts to the host device comes at the cost of slower inference than the previous approach. Techniques like parameter compression [3], [6], [9], [11], [20], expert activation prediction [18], [20], and caching [6] are used to mitigate runtime degradations.

While these techniques can help speed up the MoE inference, these alone are insufficient solutions. We profile the time of reading experts from the host CPU vs from the H100 GPU. 2 compares the read time GPU and CPU for different expert sizes. We can clearly see the CPU read time is orders

of magnitude greater than the GPU read time. This would mean prefetching experts with 1-2 layers would not result in any meaningful speedup of modern GPUs. Hence, we aim to increase the use of experts already present in the HBM memory.

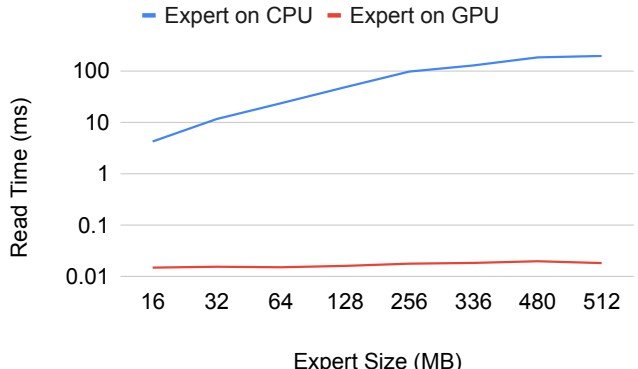

Fig. 2: Expert read time from CPU vs H100 GPU.

## III. MoE:ERAS

In this section, we first present the analysis of the expert activation patterns in Mixtral-8x7b and Switch Transformer-32 MoE models during inference. By examining the predictability and disparity in the activation patterns in this model, we present an analysis that motivates the design of MoE-ERAS. We then describe two routing schemes proposed and tested in this work - **Thresholding** and **Biasing**.

### A. Expert Activations Prediction

Mixtral-8x7b contains 32 hidden layers, with each containing 8 experts. During the generation of each token, $k$ of these experts (default k = 2) are selected by the gating network (or router) immediately prior to the layer. These experts are said to be *activated*. We hypothesize that:

- (a) despite the use of a load balancing loss during training, the activation of experts will be uneven within each layer, creating "hot" and "cold" experts that are often and rarely activated, respectively.
- (b) There exist expert-expert activation correlations between the different layers in the model, mainly between the early and late layers.
- (c) given the experts activated in earlier layers, it is possible to predict with reasonable accuracy the experts activated in later layers.

To test these hypotheses, we profile the text generation task on a corpus [16]. Inherently, the support for these hypotheses shows opportunities for optimizing the inference latency and throughput of MoE models. The support we find for these hypotheses helps us design the interventions described in Section 3.2 to the gating mechanism to speed up inference. While our work focuses on running state-of-the-art MoE models on commodity hardware where *batch size = 1* is acceptable, we

also examine how expert activation varies with batch size in Switch Transformer-32.

## B. Expert Residency Aware Routing

The behavior of the gating network in standard Mixtral-8x7b is shown below. The output of self-attention is passed through a dense network, which gives logit values for each expert. Softmax function is applied to these logits to convert them into probabilities. The Top-K method is used to select the experts to activate, where $k$ is a parameter derived from the model configuration. In Mixtral-8x7b, this defaults to $k = 2$.

$$Logits = H_i * W_{exp} \tag{1}$$

$$Weights = Softmax(Logits) \tag{2}$$

$$Experts_{Activated} = SelectTopK(Weights) \tag{3}$$

Our motivation results from section II show that offloading experts to the host can significantly affect inference latency. MoE Offloading [6] selects experts to cache and offload, making it possible to run Mixtral-8x7b on a Tesla T4 with 16GB VRAM. Our examination of the gating network's outputs shows that there are two key ways in which we can make our gating flexible -selecting "good-enough" experts and "biasing" towards those already on-chip.

*1) Thresholding:* Early analysis of the gating network's output logits showed us that there might not always be "clear winners" when selecting experts. Sometimes, an expert is selected because it is marginally better than other close competitors. If it so happens that the top expert is off-chip, this represents an opportunity to loosen the bottleneck in a memory-bound decoding process.

The **thresholding** approach aims to select good-enough experts by boosting the activation probability of on-chip experts artificially by $\alpha$, a user-defined hyper-parameter. This has the effect of tipping the balance in favor of the experts on-chip in cases where a close competitor to the top expert is on-chip. The equation below describes how the probability is adjusted for an expert $E_i$, in a *residency-aware* manner.

$$Weights_i = \begin{cases} Weights_i + \alpha & \text{if } E_i \text{ is in fast mem} \\ Weights_i & \text{if } E_i \text{ is in slow mem} \end{cases}$$

*2) Biasing:* Using the profiling defined in subsection III-A, we estimate the normalized activation frequencies $freq$, as shown in Figure 3a. We then define a more expressive penalty for off-chip experts that penalizes the choice of experts by both the frequency of its activation and scales it by the user-defined hyper-parameter $\beta$. The key idea in the **biasing** method is that bringing a rarely used expert on-chip will likely result in it being swapped out again in a later token, creating two swaps between HBM and host memory in an already memory-bound process. Instead, settling for a competitor that is likely to be reused improves latency. This is an accuracy-performance trade-off controlled by the user through $\beta$, but it also considers the fact that the frequently used experts are likely to be the top choice for later tokens, which presents a second advantage of biasing over thresholding.

The equation below describes the use of the penalty to adjust the logit for expert $E_i$ in cases where it is on or off-chip. Note that in contrast to thresholding, we adjust the raw logits and then apply the softmax function to obtain the final probabilities.

$$Logits_i = \begin{cases} Logits_i & \text{if } E_i \text{ is in fast mem} \\ Logits_i - \beta(1 - freq(E_i)) & \text{if } E_i \text{ is in slow mem} \end{cases}$$

## IV. EVALUATION

In this section, we verify our earlier hypotheses about MoE behavior and benchmark the inference latency with different configurations. We focus our quality evaluations on Mixtral-8x7B models since they represent the current state of the art among open-access MoE models. We organize this section as follows: subsection IV-A presents the key insights for expert activations based on the hypothesis presented in subsection III-A. subsection IV-B compares the real system speedups when using MoE-ERAS. Finally, subsection IV-C measures the quality implications of using the resident expert.

## A. Expert Activation Patterns

We examined the hypotheses in subsection III-A by running a large inference workload on a text summarization task using Mixtral-8x7b, and Switch-T-32E on the CNN DailyMail Dataset [16]. It is a summarization dataset that contains long text *articles* and condensed summaries of the article called *highlights*. This allows the collection of the activations in both the pre-fill and decode stages, but we focus on the sequential decode phase for analysis. We collect output logits from the gating networks and infer the $k$ selected experts and the softmax distribution in each stage. For Mixtral, given $h = 32$ hidden layers, each containing $E = 8$ experts, we obtain a $32 \times 8$ tensor containing the logits from gating networks. For Switch Transformer, this is instead $h = 6$ and $E = 32$, giving a $6 \times 32$ activation tensor.

We collect activation data over 139k tokens for Mixtral and 500k tokens for Switch Transformer. The analysis of the router activations presents support for our hypotheses. Figure 3a and Figure 3b present distributions normalized along each layer. The dark and light spots in these visualizations represent experts that are rarely and frequently activated, respectively, where activation is defined as being in the top $k = 2$ experts within the layer. This supports our hypothesis (a). For Mixtral, a perfectly equitable distribution of tokens would give 0.125 for all experts, and the visualization confirms that many experts are above that threshold. Likewise, for the Switch Transformer, many experts are activated well in excess of 0.03 (1/32).

In order to study expert-expert activations, we obtain the output logits from each gating network (32 for Mixtral, 6 for Switch Transformer), apply the softmax function, and build a correlation matrix. The correlation matrix shows both cells with high positive and negative correlations. Considering that

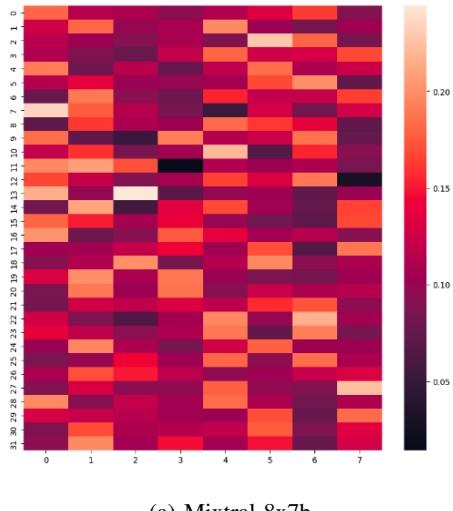

(a) Mixtral-8x7b

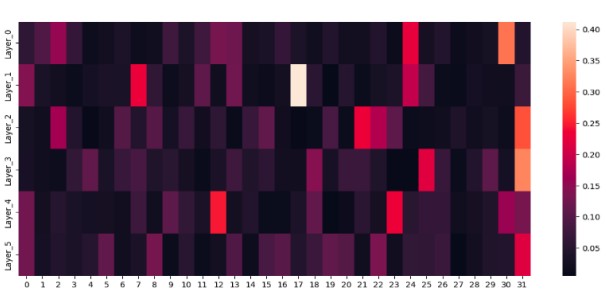

(b) Switch Transformer-32E

Fig. 3: Activation patterns for different MoE models.The lighter cells indicate high activation frequency, while the darker cells correspond to rarely activated experts.

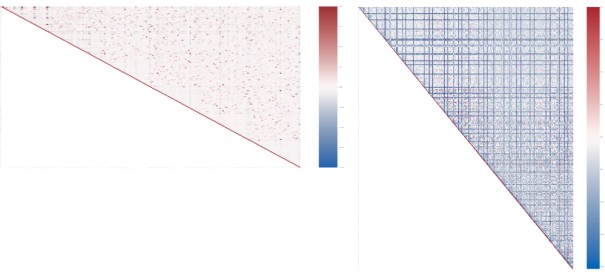

Fig. 4: Correlation coefficient between gating network outputs logits across all layers for Switch Transformer (left) and Mixtral-8x7b (right). The plot shows dark-red and dark-blue spots, representing high positive and negative correlations. These represent opportunities to potentially pre-fetch experts based on these probabilities.

this was collected over 500k tokens, this supports the idea that expert-expert correlations of reasonable strength exist regardless of the input tokens.

This leads us to the idea that we might be able to see greater predictability by considering the first few layers together. If it is possible to predict the expert activations of layers deep in the network, then it can support scheduling expert pre-fetching for requests ahead of time. Predicting expert activation based on the first few layers is particularly meaningful since this can lend itself to efficient pre-fetching. We use the activation probabilities of each expert in the first 4 layers as the input features to the regression model. We predict the activated experts for $k = 2$ in layers 8 - 32, and the prediction accuracy scores are shown in Figure 5a. For the Switch Transformer, Figure 5b shows an attempt to predict softmax values instead of activations. The promising results show that we consistently beat random chance significantly. With Mixtral, we are able to predict the activated experts in first and second place with ¿50% accuracy in all experts, with some surpassing 70%. These beat the random chance of 12.5% substantially.

There are three important factors to weigh while considering these results. First, this may be an underestimation of the accuracy since we individually predict the first and second experts. If the regression selects the same two experts in inverse order, it diminishes the accuracy but has no effect for the purpose of pre-fetching. Second, this is simple linear regression, and the intention is to show that they are correlated simply. In implementing this in a scheduler, it may be replaced with a few fully connected layers with non-linearity to improve prediction accuracy. Finally, the pattern is borne out over our workload of 500k tokens, and while we believe this to be a general pattern, distribution shifts may be addressed by periodically re-calibrating the regression as done in other works such as MoE-Infinity [**?**].

Similar interpretations are borne out for the Switch Transformer. There, we try to predict the softmax values of the experts directly - a harder problem. We see that we still beat random chance consistently, as shown in Figure 5b.

However, key challenges remain to be explored in this direction. This paper focuses on cases where $batch\_size = 1$ is applicable, such as commodity and edge settings. However, we also analyze the growth of the number of activated experts with batch size (without thresholding/biasing) to understand the applicability of this work to resource-constrained settings where larger batch sizes are desired. Figure 6 shows this analysis for Switch Transformer-32E, and the number of distinct experts grows to about $E/2$ at batch size 16. The later layers also consistently show more diversity than the earlier layers, which is an interesting perspective. This supports the idea that pre-fetching the later layers using the predictions presented above can present substantial gains in future improvements of this work.

### B. ERAS - Speedups

In this section, we profile the speed-ups we are able to achieve with biasing and thresholding. We examine the se-

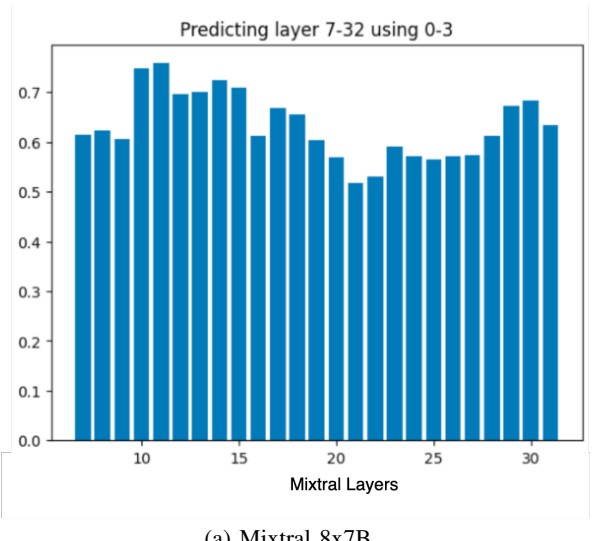

(a) Mixtral 8x7B

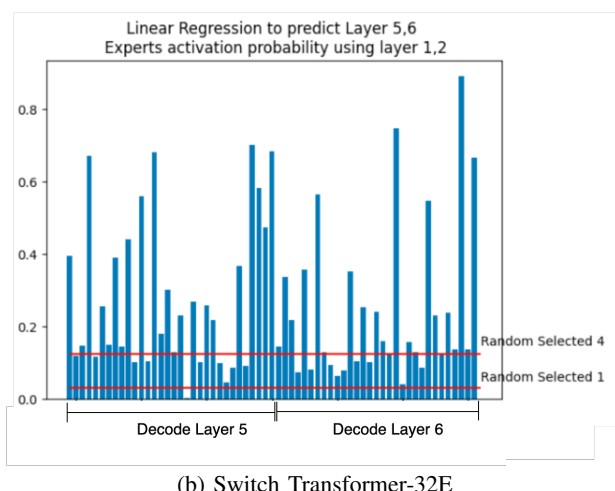

(b) Switch Transformer-32E

Fig. 5: Accuracies of predicting later layer expert activations using early layer's logit values.

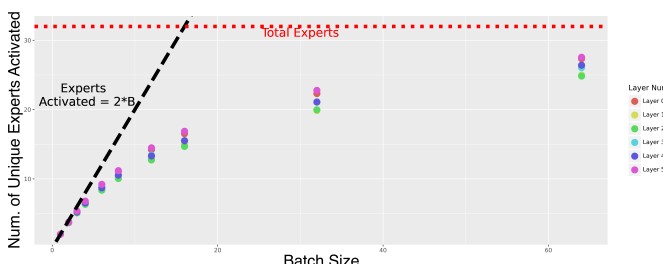

Fig. 6: Number of *distinct* experts activated as the batch size grows for switch-T-32E. Unique experts are growing sublinearly compared to 2×Batch
.

quential decoding to count the number of expert loads saved, and the overall impact on latency. We compare against the baseline implemented in dvmazur/mixtral-offloading, which includes quantization and expert caching. Our optimizations are **orthogonal** to these and can be applied with or without other techniques. We consider our top-K routing with quantization and LRU caching as proposed in Moe-offload as our baseline. We generate sequences with $l = 100$ tokens, $n_{iter} = 50$ times, and collect the mean latency (wall clock time) and throughput. We see substantial gains using these approaches as seen in Figure Figure 7.

We see two insights from this result:

First, the threshold determines the savings. In all offload settings, thresholding requires selecting $\alpha$, which we test at $0.05, 0.15, 0.25$. We find that as the threshold increases, the performance improves owing to saving more expert offloads. Together with the quality metrics in subsection IV-C, a threshold can be selected for the desired performance. These performance metrics should only be compared between approaches in this paper, as latency, throughput, and tokens/second metrics are hardware-dependent. While the ordering should be the same on other hardware, the actual numbers will likely differ.

Second, as $offload\_per\_layer$ grows, the savings become more significant. As more experts are offloaded when less VRAM is available, it becomes more likely that an off-chip expert is activated, causing performance degradation while the decoding waits for experts to be brought into memory. This shows that as the environment gets more and more resource-constrained, our approach becomes more important.

In summary, depending on the number of experts offloaded, we find that we can achieve 10% - 13% reduction in latency using thresholding at $\alpha = 0.15$, and 8.0% to 9.7% reduction using biasing with $\beta = 1$. At higher $\alpha$, we achieve even more savings as shown in Figure 7. Since this work represents a performance-accuracy trade-off, subsection IV-C examines the quality of the generation with these performance gains.

### C. ERAS - Quality

Next, we test how different residency-aware routing schemes affect MoE inference quality. We only perform the quality experiments with Mixtral-8x7B as that is the SOTA MoE open-source model. We measure perplexity for Wiki-Text2 [13] and C4 [15]. We also measure 5-shot MMLU [8] accuracy. For WikiText2 and C4, we use the test set and validation sets, respectively. We use a sliding-window strategy with a stride of 512 and a max generation length of 2048. For MMLU, we ran the test over the complete dataset.

As shown in Table I, our expert activation technique presents minimal quality degradation at low threshold values. As we increase the threshold $\beta$, the quality goes down. This result and speedup seen in the previous section present a quality-speedup trade-off for MoE model inference.

### V. RELATED WORKS

Several prior efforts have a similar goal of reducing the inference latency of mixture-of-experts models.

EdgeMoE [20] aims to reduce the latency of inference of MoEs on edge systems. It uses quantization and 1-2 layer early

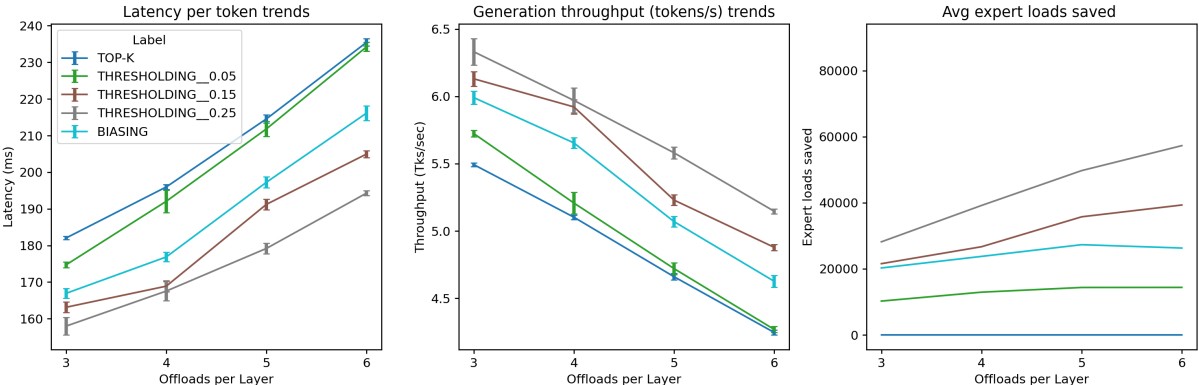

Fig. 7: Speedup provided by various offloading algorithms over top-k routing. *Our baseline top-K routing already has implemented different optimizations like parameter quantization and LRU caching*. Varying $\alpha$ and offload_per_layer shown.

| Method | C4-PPL | WikiText2-PPL | MMLU-Acc. |
|---|---|---|---|
| Top-K | 8.044 | 4.497 | 66.1 |
| THRES-0.05 | 8.062 | 4.512 | 66.1 |
| THRES-0.10 | 8.133 | 4.560 | 66.1 |
| THRES-0.15 | 8.221 | 4.625 | 66.1 |
| BAISING | 8.300 | 4.679 | 66.1 |
| THRES-0.25 | 8.522 | 4.813 | 66.1 |

TABLE I: Quality results of different expert activation techniques on different datasets.

expert prediction to fetch which experts would be activated appropriately. All non-expert weights are kept on the chip. However, this work is aimed at edge devices like Raspberry Pi and might not work on GPUs. Pre-gated MoE [10] changed the model architecture to predict the experts one layer early. Expert Affinity [19] provides a solution for a multi-GPU setup with expert parallelism. They propose method to reduce cross-GPU communication using KV cache duplication. Using this technique, they propose having 1 A2A + 1 AG instead of 2 A2As during inference. MoE-Infinity [18] performs activation-aware prefetching and caching of experts. They use a sample workload (e.g., validation) to form Expert Activation Matrices (EAMs) that they store in a collection. They rely on temporal locality (repeated activation of an expert in a sequence) and sparse activation (only a few activated) assumptions to select the expert to cache and prioritize the prefetch. MoE-offload [6] propose quantization along with LRU caching and hidden state-based expert prediction for MoE inference on commodity hardware. While all these works are focused on expert pre-fetching and/or quantization, our work focuses on taking expert residency into account. Thus our work is orthogonal to all the related works and can be implemented along with any other proposed quantization or prefetching technique.

## VI. CONCLUSION

In this paper, we have shown that Expert Residency Aware Selection (ERAS) shows considerable performance gain for those running Mixtral-8x7b in resource constrained environments requiring expert offloading to host memory. We provide parameters the user can tune navigate the accuracy-performance trade-off, and show that the impact of ERAS

on perplexity or accuracy is minimal compared to the performance benefit it offers. This can be applied on top of, or instead of other approaches like parameter compression for performance gains.

However, this work comes with limitations. While our profiling and analysis include Switch Transformer as well, our implementation is limited to Mixtral at the moment. While we show both downstream tasks and text generation accuracy, a larger validation on all available benchmarks is required to establish the accuracy retained on other tasks. In addition, since both thresholding and biasing are inference time changes, they may redirect tokens to experts that have seen few such training examples, leading to increased risk of hallucinations.

In our next steps, we aim to establish test it on more comprehensive evaluation benchmarks, implement it for other MoE models, and study the effect of biasing/thresholding without aggressive quantization to compare the trade-offs.

## VII. ACKNOWLEDGMENT

We would like to thank Ganesh Murugappan and Dr. Anand Iyer for their assistance during the initial brainstorming phase. We also extend our gratitude to PACE@Georgia Tech and CRNCH lab for providing the necessary GPUs.

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
