# OpenReview forum: "MoE-ERAS: Expert Residency Aware Selection"
_iscaconf.org/ISCA/2024/Workshop/MLArchSys — MLArchSys 2024 OralPoster_

### Official Review · Reviewer_9fD5 · 2024-05-27
**This work seems like a good start to efficient MoE Execution**

**Confidence:** 4
**Rating:** 6

**Detailed Feedback And Questions For Authors:**

The authors are presenting their technique to improve the performance of MoEs by predicting routing and by factoring the expected residency of the experts.  This work is finding an interesting way to deal with resource scarcity of MoE execution.  While overall the work seems to be moving in the right direction, there are some areas where it can be improved.

The primary issue I would like to see the authors address is looking at the performance versus accuracy and the cost or timing of determining the expert routes.  I believe there is still some techniques to be explored here and the costs evaluated in a way that is more efficient besides just predicting using another network.  There may be some training changes that make it easier to allocated to multiple experts at low cost.

Overall, I would like to see how doing this may impact the hardware.  Is there something we can add to the hardware that will improve over performance when using the MoE allocation scheme proposed here?  Basically, I would like more hardware and the impact on hardware from this work.

**Top Reasons To Accept The Paper:**

MoE is becoming more popular for LLMs and better performance is required so this work is timely.
The authors are tackling the problem in an interesting way.
There is a few ways the authors can move forward with the work.

**Top Reasons To Reject The Paper:**

Some components about accuracy versus performance need further exploration.

---

### Official Review · Reviewer_eNEB · 2024-05-28
**Expert selection based on heuristics for efficient MoE models**

**Confidence:** 3
**Rating:** 7

**Detailed Feedback And Questions For Authors:**

The paper proposes heuristic-based techniques to make MoE models more efficient for inference. The proposed techniques, thresholding and biasing, can be orthogonally applied with other optimizations proposed for MoE models. The authors profile the Mistrail and SwiftTransformer models to gather heuristics on determining which experts get activated based on correlation. That allows reduced movement of expert parameters from HBM to low speed memory. All ideas are well discussed and motivated. The evaluations are very detailed and insightful. The paper's writing could be further improved.

1. There are some grammatical issues in Section 2.2 and Section 4.1.

2. While the insights and existing evaluation are detailed, more evaluation on diverse benchmarks is required to see if similar insights and heuristics still hold.


3. Are both thresholding and biasing applied together? Are there any benefits to doing that? Is biasing + thresholding included in the evaluation in Figure 7?

4. The author suggests that experts for next layers can be prefeteched based on the prediction. What are the differences and similarities with other existing prefetching techniques? Are there any tradeoffs?

**Top Reasons To Accept The Paper:**

The paper proposes novel techniques for expert selection in MoE models that minimize expert offloading to slow memory based on insights from activation patterns. The authors include detailed insights on activation patterns for Mixtral-8x7B and Switch Transformer-32E.

**Top Reasons To Reject The Paper:**

More evaluations of proposed techniques on diverse benchmarks are required for validation.

---

### Official Review · Reviewer_JDL9 · 2024-05-28
**The paper was interesting to read with simple effective approaches, however adding data for more meaningful insights could improve it a lot**

**Confidence:** 4
**Rating:** 6

**Detailed Feedback And Questions For Authors:**

## Summary

This work aims to slightly tweak the expert selection algorithm in an MOE layer to enable performance improvement by preferential selection of resident experts. To achieve this goal, this work proposes and evaluates speedups and quality across two key techniques (1) Thresholding (2) Biasing which nudge the expert probability activation to prefer resident experts. It claims that such an approach adds minimal degradation to quality.


## Few suggestions/comments:

- Profiling expert activations to show hot and cold experts is very insightful. Section 4.1 shows the data across all tokens in section 4.1. Are there any more insights on correlation between expert activation and input queries, can input query categorization help ?
- Biasing is a good technique, but it has a tendency to downplay or completely avoid a rare expert, as it is reliant on past frequency. Some added functionality to mitigate this corner case may be better. I am hinting at approaches similar to branch prediction, where recent history is accounted for prediction instead of total history. Also input categorization may have its own implications on this effectiveness of biasing.
- Figure 4 for correlation coefficient is unclear, could you add more labeling to the figure indicating what exact data is correlated ?
- Expert Prefetching:
  - Does Figure 5b shows probabilities, what is on the X-axis ?
  - The speedups from expert prefetching has not been evaluated right?
- Batching effects:
  - Could you add discussion on how the proposed ERAS techniques work in the case of increased batching ?
- Please provide information on platform/setup/methodology for profiling in 4.2, Eras speedups.
  - The github repo seems to use rely on the GPU alloted to the google colab, which seems to be a V100.
  - If possible, numbers from A100 would be more insightful considering the current GPU deployments in datacenter. As A100s or H100s have increased memory capacity from 40 GB to 80 GB memory, gains on these chips are relevant.
- Could you expand the Results in Section 4.3
  - Biasing parameter $\beta$ - sensitivity? It will be interesting to see how speedups and quality vary with $\beta$
  - Could you include an experiment with both thresholding and biasing ? Alpha and beta together?

**Top Reasons To Accept The Paper:**

- I think the paper has good motivation, goals and attempts to attain it with simple techniques. The scope for improvement is highlighted by latency gap between device memory vs host memory, and the intuition of residency is backed by a profiling of expert activation on MOEs like Mistral-8x7B and Switch Transformer-32E.

- Simple and effective approaches like thresholding, biasing and prefetching are suggested in this work

**Top Reasons To Reject The Paper:**

- The paper needs more clarity between ideas suggested vs evaluations discussed. A prefetching idea was discussed in Section 4.1, predicting experts based on data from initial layers seems to be a good idea, however performance gains from it are not evaluated.

- Experimental setup details, additional information on how latency is measured.

- Some sensitivity analysis across all parameters would be useful.

---

### Official Review · Reviewer_6h3w · 2024-05-30
**Interesting activation selection strategy, yet more cost analysis is needed show if it works and what scenario it fits better than the previous work**

**Confidence:** 3
**Rating:** 5

**Detailed Feedback And Questions For Authors:**

The paper describes the overall design of Profiling Expert Activations and Expert Residency Aware Routing. It would be better to share more insight on how to deploy the overall selection strategy into the system. It would be better if the author can address these questions: Is it one-time cost of training and building such regression model for prediction accuracy? Do we need recurring training to make it adaptive to different resources of tokens? What is the extra cost of running such regression model during inference and serving?

**Top Reasons To Accept The Paper:**

This paper proposes "Expert Residency Aware Selection", which builds a regression model from the profiling result to increase the reusablity of experts that are already present on the HBM rather than bring new experts each time from host memory.

**Top Reasons To Reject The Paper:**

The idea is interesting yet more experiment is needed to verify if it works. The result would be more trustworthy if the author can explain how the prediction accuracy will affect the e2e metrics (i.e. latency and throughput). It would be better to show what is the sweet spot when trading "extra cost of better prediction accuracy w/ more complex regression model" by "better latency & throughput".

---

### Decision · Program_Chairs · 2024-05-30

**Decision:**

Accept (Oral/Poster)

**Comment:**

Congratulations! We are pleased to inform you that your paper has been accepted for presentation at MLArchSys 2024. We look forward to your participation at the workshop. Further details regarding the schedule and format will be provided soon. See you at the workshop!